# Isolation and Identification of Bioactive Compounds from *Streptomyces actinomycinicus* PJ85 and Their In Vitro Antimicrobial Activities against Methicillin-Resistant *Staphylococcus aureus*

**DOI:** 10.3390/antibiotics11121797

**Published:** 2022-12-10

**Authors:** Panjamaphon Chanthasena, Yanling Hua, A’liyatur Rosyidah, Wasu Pathom-Aree, Wanwisa Limphirat, Nawarat Nantapong

**Affiliations:** 1School of Preclinical Sciences, Institute of Science, Suranaree University of Technology, Nakhon Ratchasima 30000, Thailand; 2The Center for Scientific and Technological Equipment, Suranaree University of Technology, Nakhon Ratchasima 30000, Thailand; 3Research Center for Vaccine and Drug, National Research and Innovation Agency (BRIN), Bogor 16911, Indonesia; 4Department of Biology, Faculty of Science, Chiang Mai University, Chiang Mai 50200, Thailand; 5Center of Excellence in Bioresources for Agriculture, Industry and Medicine, Chiang Mai University, Chiang Mai 50200, Thailand; 6Synchrotron Light Research Institute, 111 University Avenue, Nakhon Ratchasima 30000, Thailand

**Keywords:** dihomo-γ-linolenic acid, *Streptomyces* sp., *Streptomyces actinomycinicus*, antibacterial activity, drug-resistant microorganisms

## Abstract

Antibiotic-resistant strains are a global health-threatening problem. Drug-resistant microbes have compromised the control of infectious diseases. Therefore, the search for a novel class of antibiotic drugs is necessary. Streptomycetes have been described as the richest source of bioactive compounds, including antibiotics. This study was aimed to characterize the antibacterial compounds of *Streptomyces* sp. PJ85 isolated from dry dipterocarp forest soil in Northeast Thailand. The 16S rRNA gene sequence and phylogenetic analysis showed that PJ85 possessed a high similarity to *Streptomyces actinomycinicus* RCU-197^T^ of 98.90%. The PJ85 strain was shown to produce antibacterial compounds that were active against Gram-positive bacteria including methicillin-resistant *Staphylococcus aureus* (MRSA). The active compounds of PJ85 were extracted and purified using silica gel column chromatography. Two active antibacterial compounds, compound 1 and compound PJ85_F39, were purified and characterized with spectroscopy, including liquid chromatography and mass spectrometry (LC–MS). Compound 1 was identified as actinomycin D, and compound PJ85_F39 was identified as dihomo-γ-linolenic acid (DGLA). To the best of our knowledge, this is the first report of the purification and characterization of the antibacterial compounds of *S. actinomycinicus*.

## 1. Introduction

Ehrlich and Sata introduced arsphenamine (salvarsan) in 1910, and it is considered to be the first synthetic antibiotic [1]. It was widely used against syphilis and tripanosomiasis [1]. However, the first naturally occurring antibiotic, penicillin, was discovered by Alexander Fleming in 1928 [2,3]. Penicillin was successfully used to control bacterial infections during World War II [4]. Since then, many antibiotics have been discovered and administered to humans and animals for therapy and prophylaxis [5]. The misuse and widespread use of antibiotic drugs have led to the emergence of resistant pathogenic microorganisms [6,7].

Antibiotic resistance emerged in early 1942 when *Staphylococcus aureus* became resistant to penicillin [6]. In 1960, methicillin was then used to cure penicillin-resistant *S. aureus*; however, resistance to methicillin was soon developed [8]. The first case of methicillin-resistant *S. aureus* (MRSA) was detected in the United Kingdom in 1962 [6]. Nowadays, MRSA infection is a global health concern since it causes high rates of morbidity and mortality [9,10,11]. MRSA can resist almost all therapeutic β-lactams and other classes of antibiotics [12]. Vancomycin has been identified as the last drug for treating MRSA [12,13]. However, the vancomycin-intermediate *S. aureus* (VISA) strain was first reported in Japan in 1997 and later in several countries [14]. Recently, many strains, including *Klebsiella pneumoniae*, *Pseudomonas aeruginosa*, *Acinetobacter baumannii*, and *Enterobacter* spp., have adapted to resist antibiotic drugs [15]. Therefore, new potent antibiotic agents are urgently needed, particularly against antibiotic-resistant pathogens [16,17,18,19].

The filamentous actinomycetes bacteria are an important source of antibiotics. About 80% of antibiotic drugs are obtained from actinomycetes, mostly from the *Streptomyces* and *Micromonospora* genera [20,21,22,23]. However, only 1–3% of all known antibiotics produced by *Streptomyces* have been isolated [24,25]. Hence, an immense number of antimicrobial compounds remain to be discovered. This study was focused on the identification of antibacterial compounds produced by the soil isolate *Streptomyces* sp. PJ85. The antibacterial compounds of PJ85 were purified and characterized using silica gel column chromatography, TLC, and LC–MS analysis.

## 2. Results

### 2.1. Identification and Characterization of PJ85 Strain

According to a previous study, a total of 123 bacterial soil isolates were isolated from dry dipterocarp forest soil around Suranaree University of Technology, Nakhon Ratchasima, Thailand (14.8729° N, 102.0237° E) [26]. The isolates were tested for antimicrobial activity against test pathogens including Gram-positive and Gram-negative bacteria. The results showed that the PJ85 strain showed a high antibacterial activity against test pathogens. Therefore, the PJ85 strain obtained from the previously study was used in this study [26]. The PJ85 strain is Gram-positive, aerobic, and filamentous in nature. Morphological observations of a 14-day-old culture grown on an ISP-2 agar medium revealed the rich growth of aerial and vegetative hyphae. The colors of the aerial and vegetative hyphae were light and strong yellow, respectively. The PJ85 strain also produced a strong yellow diffusible pigment after incubation for 14 days at 37 °C (Appendix A available in the online Appendix A). The 16S rRNA gene of PJ85 was sequenced and compared with reference sequences from the EzBiocloud database (https://www.ezbiocloud.net, accessed on 9 November 2022). The results revealed that the 16S rRNA gene sequence of the PJ85 strain (1523 nt) was closely related to the members of the *Streptomyces* genus. The PJ85 strain shared the highest 16S rRNA gene sequence similarity with *S. actinomycinicus* RCU-197^T^ (99.86%), *Streptomyces echinatus* NBRC12763^T^ (98.74%), and *Streptomyces graminisoli* JR-19^T^ (98.48%). The sequence of the 16S rRNA gene of PJ85 was submitted to GenBank under accession number MK580459.

In the neighbor-joining phylogenetic tree based on the 16S rRNA gene sequences, PJ85 formed a clade with its closest relative strain obtained from the EzBiocloud database. The PJ85 strain shared a node with *S. actinomycinicus* RCU-197^T^, with a bootstrap value of 100% (Figure 1). Therefore, PJ85 may be closely related to *S. actinomycinicus*.

### 2.2. Antibacterial Activity of PJ85

The antibacterial activity of *Streptomyces* sp. PJ85 was tested against methicillin-resistant *Staphylococcus aureus* (MRSA) DMST20651, *Staphylococcus aureus* ATCC29213, *Staphylococcus epidermidis* TISTR518, *Bacillus subtilis* TISTR008, *Bacillus cereus* TISTR687, *Escherichia coli* TISTR780, *Enterobacter aerogenes* TISTR1540, *Pseudomonas aeruginosa* TISTR781, *Serratia marcescens* TISTR1354, *Proteus mirabilis* TISTR100, and *Salmonella typhimurium* TISTR292 using the perpendicular streak method. *Streptomyces* sp. PJ85 exhibited antibacterial activity against Gram-positive bacteria, including MRSA, while Gram-negative organisms were not affected.

The maximum value of the inhibition zones (mm ± SD) of the antibacterial activity of *Streptomyces* sp. PJ85 was found against MRSA DMST20651 (50.00 ± 0.00), followed by *S. epidermidis* TISTR518 (48.33 ± 2.89), *S. aureus* ATCC29213 (46.67 ± 0.58), *B. subtilis* TISTR008 (45.00 ± 3.00), and *B. cereus* TISTR687 (38.33 ± 1.15). The antibacterial activity of *Streptomyces* sp. PJ85 against test pathogens according to the perpendicular streak method is shown in Table 1. According to our results, the zone of inhibition of the antibacterial activity of PJ85 against MRSA DMST20651 was significantly higher than that of *B. subtilis* TISTR008 and *B. cereus* TISTR687 (*p* < 0.05). Although the antibacterial activity of PJ85 against MRSA DMST20651 was not statistically different from that of *S. aureus* ATCC29213 and *S. epidermidis* TISTR518 (*p* > 0.05), the PJ85 strain exhibited a larger zone of inhibition than *S. aureus* and *S. epidermidis* (Table 1).

### 2.3. Incubation Temperature, Period Affect Growth, and Antibacterial Activity of PJ85

The cultural temperature for the growth and antibacterial activity of PJ85 was studied on an ISP-2 medium because the PJ85 culture on ISP-2 exhibited the highest antibacterial activity compared with other media (data not shown). In order to study the effect of incubation temperature on growth and antibacterial activity, PJ85 was incubated at two temperatures, 30 °C and 37 °C. The fermented broth of PJ85 was collected every day for 14 days. The cell biomass and cell-free supernatant were separated by filtration. The cell pellet of PJ85 was dried to obtain the dry cell weight, while a cell-free supernatant was used to prepare a crude extract of ethyl acetate. The results showed that the maximum growth was observed at 30 °C on day 4 of cultivation, with a biomass yield of 4.40 ± 0.25 mg/mL (Figure 2). There was a statistically significant difference between incubation temperature (30 °C and 37 °C) and incubation period on the growth of PJ85 (Figure 2; *p* < 0.0001). On individual days of incubation, the cell biomass of PJ85 cultured at 30 °C was significantly higher than that cultured at 37 °C on days 4, 5, 6, 7, 8, 10, and 11 (Figure 2). The extracts were then used to determine antibacterial activity using the disc diffusion method. The results revealed that the antibacterial activity of the PJ85 crude extract against Gram-positive bacteria was highest around day 5. The antibacterial activity was stable from days 5 to 9 of cultivation and began to decrease at day 10 (Figure 3). A two-way ANOVA with post hoc Bonferroni correction revealed a significant difference between incubation temperature (30 °C and 37 °C) and time on the antibacterial activity of the crude extract of PJ85 against MRSA DMST20651, *S. aureus* ATCC29213, *B. subtilis* TISTR008, and *B. cereus* TISTR687 (Figure 3; *p* < 0.0001). The antibacterial activity against MRSA DMST20651, *S. aureus* ATCC29213, and *S. epidermidis* TISTR518 of the crude extract of PJ85 cultivated at 37 °C was significantly higher than that at 30 °C on day 1 (Figure 3A–C). Bonferroni multiple comparisons showed that the crude extract of PJ85 grown at 37 °C had a significantly higher antibacterial activity than that at 30 °C against *B. subtilis* TISTR008 (days 5–10; Figure 3D) and *B. cereus* TISTR687 (days 1–3 and days 10–14; Figure 3E).

According to Student’s *t*-test (Table 2), the antibacterial activity of the PJ85 crude extract obtained from cells grown at 37 °C possessed a significantly larger zone of inhibition against *B. subtilis* TISTR008 and *B. cereus* TISTR687 than that grown at 30 °C (two-tailed *t*-test, *p* < 0.05). Although the antibacterial activity against MRSA DMST20651, *S. aureus* ATCC29213, and *S. epidermidis* TISTR518 was not found to be statistically different at both cultural temperatures, the activities were somewhat higher when the cells were cultured at 37 °C than at 30 °C. Thus, a cultivation temperature of 37 °C and incubation period of 5 days were applied for the preparation of crude extracts in order to obtain the maximal yield of antibacterial activity.

### 2.4. Crude Compound Preparation and MIC Values

The cultural conditions of PJ85 that yielded the highest antibacterial activity were used to prepare the crude compounds. Therefore, PJ85 was inoculated in an ISP-2 medium (200 mL in a 1000 mL Erlenmeyer flask without baffled) and incubated at 37 °C and 200 rpm for 5 days. After incubation, the cell-free supernatant was collected to extract the crude compounds. In order to extract the crude compounds of PJ85, different solvents such as n-hexane, n-butanol, chloroform, ethyl acetate, ethanol, and methanol were tested. The ethyl acetate crude extract showed the highest antibacterial activity of the tested solvents (data not shown). Therefore, ethyl acetate was used for the preparation of the crude compounds. The crude compounds of PJ85 were yellowish-orange in color. The yield of the crude compounds was 246.54 ± 17.12 mg/g of dry cell weight.

The yellowish-orange crude ethyl acetate extract of PJ85 was used for an evaluation of MIC by using the two-fold macro-dilution method. The MIC values of the crude ethyl acetate of PJ85 against MRSA DMST20651, *S. aureus* ATCC29213, *S. epidermidis* TISTR518, *B. subtilis* TISTR008, and *B. cereus* TISTR687 were 2, 2, 16, 2, and 1 µg/mL, respectively (Table 3). The assay was carried out in triplicate, during in which the same MIC values were attained.

### 2.5. Purification of the Active Compounds of Streptomyces sp. PJ85 with Thin-Layer Chromatography, Column Chromatography, and Bioautography Analysis

The separation of the antibacterial metabolites present in yellowish-orange crude compounds was performed with TLC. The mobile phase used to develop the plate was chloroform:n-hexane (9.5:0.5, *v*/*v*). After running, the TLC plates were dried and used to detect active bands on chromatogram using contact bioautography. This assay has been successfully used to determine active spots with an inhibitory effect on microbial growth [27,28,29].

Bioautography revealed two active bands, compound 1 and compound 2, on the TLC plate. These compounds exhibited antibacterial activity against Gram-positive bacteria, including MRSA (Figure 4). Based on LC–MS analysis, compound 1 was identified as actinomycin D (Appendix A). Actinomycin D is a secondary metabolite produced by many streptomycetes, including *S. actinomycinicus* RCU-197^T^, the closest related strain of PJ85 (Figure 1). It has been reported that the sole antibacterial agent produced by *S. actinomycinicus* RCU-197^T^ is actinomycin D [30]. However, an unidentified bioactive agent, compound 2, was also detected from the PJ85 crude extract (Figure 4).

In order to identify compound 2, the band consistent with compound 2 was carefully scrapped and used for purification with silica gel column chromatography. About 3.0 mg of compound 2 scrapped from 25 TLC plates was chromatographed on a silica gel column and eluted with a stepwise solvent system consisting of chloroform:n-hexane. A total of 121 fractions of 5 mL each were collected and concentrated. All fractions were tested for antibacterial activity with the agar well diffusion method. Then, the fractions showing antibacterial activity against MRSA were analyzed with TLC and bioautography. The fractions exhibiting anti-MRSA (Rf 0.68) were pooled and designated as compound PJ85_F39.

### 2.6. Liquid Chromatography–Mass Spectrometry (LC–MS) Analysis

LC–MS was used to identify compound PJ85_F39 of *Streptomyces* sp. PJ85. Based on LC–MS analysis, the ESI–MS spectra showed one major peak at *m*/*z* 307.2172 [M + H]^+^, leading to a monoisotopic mass of 306.2172 g/mol (Figure 5). The peak was analyzed and identified by matching the mass spectra with the MassBank Europe Mass Spectral Data Base (https://massbank.eu/MassBank/Search, accessed on 26 November 2022). According to a MassBank library search, the peak at *m*/*z* 307.2172 [M + H]^+^ was matched to dihomo-γ-linolenic acid, epigallocatechin, 2,3-*trans*-3,4-*cis*-leucocyanidin, eremofortin A, fenazaquin, feruloyl agmatine, fluconazole, and koumine. A summarization of the molecular weights and nearest compounds hit for the peak is shown in Table 4. However, none of these compounds have been documented as antibacterial agents except for dihomo-γ-linolenic acid [31,32,33]. The data on the molecular weight and antibacterial activity of compound PJ85_F39 provided evidence that this compound could be dihomo-γ-linolenic acid (DGLA). 

## 3. Discussion

The emergence of antibiotic resistance has been recognized as a worldwide health-threatening problem. Therefore, there is a need for novel therapeutics to replace ineffective antimicrobial drugs. Natural products are the main source of antimicrobial agents, most of which are produced by *Streptomyces* [6,41,42,43]. In this study, an antibacterial-producing *Streptomyces* PJ85 was isolated from forest soil in Nakhon Ratchasima, Thailand [26]. Molecular analysis was used to identify the PJ85 strain. The identification of microorganisms with molecular techniques has many advantages over others, such as being rapid, less laborious, sensitive, specific, and efficient [44,45,46,47,48]. Based on the study of the 16S rRNA gene sequence and phylogenetic relationship, PJ85 was found to belong to the same clade as *S. actinomycinicus* RCU-197^T^ (JCM 30864^T^). The full-length 16S rRNA gene of PJ85 shared a 99.86% sequence similarity with *S. actinomycinicus* RCU-197^T^, indicating that they belong to the same species. 

Although 16S rRNA gene sequences are conventionally analyzed in bacterial systematics, their resolution may not be sufficient for species identification [49]. The multilocus sequence analysis (MLSA) of core housekeeping genes is recognized as a powerful tool for species identification in bacteria including the *Streptomyces* genus [49]. Gene sequences of *gyr*B (DNA gyrase beta subunit), *rpo*A (RNA polymerase alpha subunit), *atp*D (ATP synthase subunit b), and *rpo*B (DNA-directed RNA polymerase subunit beta) are generally used in MLSA for *Streptomyces* species [49,50]. In this study, the sequences of *gyr*B, *rpo*A, *atp*D, and *rpo*B of PJ85 and *S. actinomycinicus* RCU-197^T^ were compared (Appendix A). The results showed a high degree of identity ranging from 98.99% to 99.58% (*gyr*B: 98.99%; *rpo*A: 99.51%; *atp*D: 99.58%; and *rpo*B: 99.08%). The similarity of these genes of PJ85 and *S. actinomycinicus* exceeded the traditionally accepted threshold of 97% for species identification [51,52]. The high degree of 16S rDNA similarity and conservation of four key housekeeping genes prompted us to classify PJ85 as *S. actinomycinicus*.

*S. actinomycinicus* RCU-197^T^ was firstly discovered in 2016 [30]. It was isolated from a soil sample of forest in Rayong, Thailand. Based on the perpendicular-streak method, the RCU-197^T^ strain was found to be active against *Micrococcus luteus*, *S. aureus*, *B. subtilis*, *E. coli*, *P. aeruginosa*, and *Candida albicans* (unpublished results). To date, there have no available reports regarding the isolation and characterization of antibacterial agents of *S. actinomycinicus*. In this study, *S. actinomycinicus* PJ85 was isolated and tested for antibacterial activity against Gram-positive and Gram-negative bacteria. The PJ85 strain showed antibacterial activity against MRSA DMST20651, *S. aureus* ATCC29213, *S. epidermidis* TISTR518, *B. subtilis* TISTR008, and *B. cereus* TISTR687. Our results revealed that the antibacterial activity of PJ85 cultured at 37 °C was higher than that at 30 °C. The exhibition of a higher antibacterial activity of PJ85 at a slightly elevated temperature might be useful in industrial processes. The advantages of industrial fermentation under high temperatures include faster reaction times and reduced cooling costs for large-scale fermentation [53]. Thus, PJ85 could be a potential candidate for the production of low-cost industrial antibiotics.

The MIC value of crude compounds of PJ85 against tested pathogens was evaluated with the broth dilution method. The results indicated that the MIC of the crude compounds ranged from 1 to 16 µg/mL. The lowest MIC value of 1 µg/mL was observed against *B. cereus* TISTR687, while the highest MIC value of 16 µg/mL was observed against *S. epidermidis* TISTR518. The crude compounds of PJ85 also inhibited the growth of MRSA DMST20651, *S. aureus* ATCC29213, and *B. subtilis* TISTR008 with an MIC value of 2 µg/mL.

The genus of *Streptomyces* has been shown to produce several secondary bioactive metabolites possessing antibacterial activity. Substantial reports are associated with the study of LC–MS for the chemical analysis of *Streptomyces* spp. [28,54,55,56,57,58]. For example, Awla et al. (2016) identified several antifungal agents such as ergotamine, amicoumacin, fungichromin, rapamycin, and N-acetyl-D, L-phenylalanine produced from *Streptomyces* sp. UPMRS4 by using LC–MS [59]. Bibi et al. (2017) reported the presence of different active compounds, including sulfamonomethoxine, sulfadiazine, ibuprofen, and metronidazole-OH, in culture extracts of *Streptomyces* sp. EA85 based on LC–MS analysis [60]. In this study, active compounds present in the ethyl acetate extract of *Streptomyces* sp. PJ85 were also identified by using LC–MS analysis. The results revealed that PJ85 produced two active compounds, actinomycin D and an unidentified compound PJ85_F39.

It has been shown that *S. actinomycinicus* RCU-197^T^, the closest related strain of PJ85, only produces actinomycin D [30]. Actinomycin D is one of the oldest chemotherapy drugs, and it has been used as an anti-tumor drug to treat childhood rhabdomyosarcoma and Wilms’ tumor [61,62]. Similar to *S. actinomycinicus* RCU-197^T^, PJ85 was found to produce actinomycin D as a major product. The antibacterial activity of actinomycin D against MRSA was previously reported by Khieu et al. with an MIC value of 0.04 µg/mL [63].

Moreover, PJ85 was able to generate a second active compound, compound PJ85_F39, that exhibited strong antibacterial activity against MRSA (Figure 4). We were able to identify the PJ85_F39 compound as dihomo-γ-linolenic acid (DGLA). DGLA is a polyunsaturated fatty acid that plays an essential role as a precursor for the biosynthesis of arachidonic acid (ARA) [31]. It was reported to exhibit many activities, such as antimicrobial, anti-inflammatory, and antiallergic activities [31]. In 2013, Desbois and Lawlor reported the antibacterial activity of DGLA against Gram-positive bacteria such as *Propionibacterium acnes* and *S. aureus* with MIC values of 128 mg/L and 1024 mg/L, respectively [32]. DGLA has been reported to have antibacterial action, although there is no evidence of its anti-MRSA activity. Thus, this is the first study to reveal that DGLA is also effective against drug-resistant microorganisms.

Previously, it has been recognized that soil fungi belonging to the *Mortierella* genus, such as *M. alpina*, *M. clonocystis*, *M. elongate*, *M. gamsii*, *M. humilis*, *M. macrocystis*, and *M. globulifera*, are effective producers of DGLA [33]. However, there have been no reports of DGLA produced by the *Streptomyces* genus. This study demonstrates the first isolation and identification of DGLA from *S. actinomycinicus*. Thus, *S. actinomycinicus* PJ85 may be a potent source of DGLA since the natural source of DGLA is limited. Moreover, the production of DGLA through *S. actinomycinicus* PJ85 could play an important role in the pharmaceutical industry.

## 4. Materials and Methods

### 4.1. Isolation, Cultivation, and Maintenance of PJ85

The PJ85 strain was isolated from forest soil collected in Nakhon Ratchasima, Thailand [26]. It was isolated with the soil dilution plate technique using starch casein agar (SCA) [48,63,64,65]. The composition of SCA was as follows: 10 g/L of soluble starch, 0.3 g/L of casein, 2 g/L of KNO_3_, 2 g/L of NaCl, 2 g/L of K_2_HPO_4_, 0.05 g/L of MgSO_4_, 0.02 g/L of CaCO_3_, 0.01 g/L of FeSO_4_, and 15 g/L of agar. The pH was adjusted to 7.2 before autoclaving. The PJ85 strain was sub-cultured on International *Streptomyces* Project medium no. 2 (ISP-2): (4 g/L of yeast extract, 10 g/L of malt extract, 4 g/L of glucose, 15 g/L of agar, and pH 7.2) [47,66,67] and maintained with a glycerol (15% *v*/*v*)-based liquid medium at −80 °C.

### 4.2. Cultural and Morphological Characteristics

The cultural morphology of PJ85 was determined on the ISP-2 medium. Morphological characteristics, such as the aerial-mass color, substrate mycelial pigmentation, and diffusible pigment production, were observed.

### 4.3. Amplification and Sequencing of the 16S rRNA Gene

Universal primers, 27F (5′-AGAGTTTGATCCTGGCTCAG-3′) and 1525R (5′-AAAGGAGGTGATCCAGCC-3′), were used for the PCR amplification of 16S rDNA of PJ85 [41]. Amplification was conducted in a thermal cycler (Thermo Scientific, Waltham, MA, USA). The PCR reaction conditions were initial denaturation at 95 °C for 5 min followed by 30 cycles of denaturation at 95 °C for 60 s, annealing at 55 °C for 60 s, and extension at 72 °C for 60 s. A final extension was conducted at 72 °C for 7 min. The amplicons were purified from 0.8% agarose gel using a FavorPrep™ GEL/PCR Purification Kit (FAVORGEN, Pingtung City, Taiwan). The purified PCR product was ligated to the terminal transferase activity (TA) cloning vector, and the recombinant plasmid was transformed into *Escherichia coli* JM109. The recombinant plasmid harboring 16S rDNA was extracted and purified with the FavorPrep™ Plasmid DNA Extraction Kit (FAVORGEN, Taiwan). The purified product was submitted for Sanger sequencing at Macrogen, Korea.

### 4.4. Phylogenetic Tree Analysis

The 16S rDNA sequence of PJ85 was compared with the EzBiocloud database (https://www.ezbiocloud.net, accessed on 9 November 2022). CLUSTAL W was used to align the 16S rRNA gene of PJ85 with its closely-related species. The neighbor-joining method was applied for phylogenetic analyses using Molecular Evolutionary Genetics Analysis (MEGA) version 10.0 software. The confidence level of each branch (1000 replications) was tested with bootstrap analysis [68]. The EzTaxon-e server (https://www.ezbiocloud.net, accessed on 9 November 2022) was used to determine sequence similarities.

### 4.5. Antibacterial Assay

#### 4.5.1. Bacterial Strains

The test organisms used in the experiments were obtained from the American Type Culture Collection (ATCC), Department of Medical Sciences Thailand (DMST), and Thailand Institute of Scientific and Technological Research (TISTR). These included *S. aureus* ATCC29213, methicillin-resistant *S. aureus* (MRSA) DMST20651, *S. epidermidis* TISTR518, *B. subtilis* TISTR008, *B. cereus* TISTR687, *E. coli* TISTR780, *E. aerogenes* TISTR1540, *P. aeruginosa* TISTR781, *S. marcescens* TISTR1354, *P. mirabilis* TISTR100, and *S. typhimurium* TISTR292. All test microorganism inocula were cultured in Mueller Hinton broth (MHB) (Himedia, Thane West, India) at 37 °C for 18–24 h.

#### 4.5.2. Perpendicular Cross Streak Method

The perpendicular cross streak method was used to determine the antibacterial activity of PJ85 [48,69,70]. The PJ85 strain was inoculated as a straight line on one side of the MHA medium. The plates were then incubated at 37 °C for 5 days in order to allow the antibacterial agents produced by PJ85 to diffuse into an agar. After incubation, test pathogens were perpendicularly crossed (T-streak) to the line of the PJ85 colony. The plates were then incubated at 37 °C for 24 h. Antibacterial activity was measured based on the distance of inhibition between the colony margin of PJ85 and the test microorganisms.

#### 4.5.3. Preparation of Crude Compounds of PJ85

In order to prepare crude compounds, the PJ85 isolate was grown at 37 °C and 200 rpm for 5 days. After incubation, the culture was filtered through Whatman No.1 filter paper (Whatman^TM^, Maidstone, UK). The fermented broth containing antibacterial was mixed with a solvent such as ethyl acetate. The ethyl acetate layer was collected and concentrated using a rotary evaporator under reduced pressure at 45 °C. The crude compounds were obtained via freeze-drying and used for the evaluation of MIC and purification.

#### 4.5.4. Disc Diffusion Method

The antibacterial activity of the crude compounds was tested using the standard disc diffusion method [71,72]. The filter paper disks (6 mm in diameter) containing crude compounds at 50 μg/disc were placed on the MHA lawn with test microorganisms (0.5 McFarland standard). The antibacterial activity was determined by measuring the size of the inhibition zone in millimeters after incubation at 37 °C for 24 h.

#### 4.5.5. Determination of the Minimum Inhibitory Concentration (MIC) of Crude Compounds of PJ85

The determination of the MIC value of the crude compounds of PJ85 was performed using the dilution method [73]. An inoculum of test pathogens in the mid-log phase was transferred to a series of tubes containing serial two-fold dilutions of the crude compounds in a liquid medium (256, 128, 64, 32, 16, 8, 4, 2, 1, 0.5, 0.25, 0.125, 0.0625, and 0.03125 µg/mL). The bacterial suspensions were added to each tube to yield approximately 5.0 × 10^5^ CFU/mL. The tubes were incubated at 37 °C for 16–18 h. A positive control and a solvent control (dimethyl sulfoxide: DMSO) were included. The MIC value was recorded as the lowest concentration of the crude compounds that inhibited the visible growth of the test organisms.

### 4.6. The Study of Incubation Temperature and Incubation Period on Growth and Activity of Antibacterial Agents

The PJ85 isolate was grown on ISP-2 broth and incubated at 30 °C and 37 °C, 200 rpm, for 14 days. The biomass and cell-free supernatant of the culture were harvested every day during the 14 days of incubation by filtration through Whatman No.1 filter paper (Whatman^TM^, UK). The bacterial cells were dried in a hot air oven, and the dry cell weight was recorded. The cell-free supernatants containing antibacterial compounds were tested for antibacterial activity using the disc diffusion method.

### 4.7. Purification of Active Compounds

#### 4.7.1. Purification of Antibacterial Compounds with Thin-Layer Chromatography

Thin-layer chromatography (TLC) was applied to separate bioactive compounds from the crude ethyl acetate. The crude compounds were spotted on TLC silica gel 60 F254 aluminum sheets (Merck, Darmstadt, Germany) and left for drying. The TLC plate was vertically placed in a developing tank containing chloroform and hexane (9.5:0.5 *v*/*v*). The solvent was allowed to run until it moved up to 80% of the TLC plate. The chromatogram was left to dry and visualized under UV light at 254 nm. The antibacterial compounds on the chromatogram were identified with the contact bioautography method.

#### 4.7.2. Purification of Antibacterial Compounds with Column Chromatography

The purification of the antibacterial compounds was performed by using silica gel column chromatography. The 230–400 mesh silica gel (Merck, Germany) was suspended in n-hexane to pack the column. The column consisted of a 40-cm-long corning glass tube with an internal diameter of 1.5 cm. The final size of the column was 30 cm. A sample not exceeding 5 mL was passed through the column while keeping the flow rate at 0.36 mL/min with a stepwise chloroform:n-hexane gradient solvent system (0.00–100). Fractions of 5 mL with each solvent system were collected, and all the individual fractions were tested against MRSA with the agar well diffusion method and the contact bioautography method.

#### 4.7.3. Contact Bioautography Analysis

Bioautography analysis was used to detect the antibacterial activity of the bioactive compounds separated on a chromatogram [56,57]. The chromatogram was placed over MHA seeded with test pathogens (0.5 McFarland standard) and left for 30 min to allow the compounds from the TLC sheet to diffuse into the agar medium. The MHA plate was then incubated at 37 °C for 24 h. After incubation, the bands of the antibacterial agent were indicated by the zone of inhibition on the medium. The active band was scrapped from the TLC plate and dissolved in methanol. The mixture was then centrifuged and filtered to remove the residual silica. The supernatant containing an antibacterial compound was used for characterization.

### 4.8. Identification of Active Compounds with Liquid Chromatography–Mass Spectrometry (LC–MS)

The mass spectrum of the active compounds was assessed using LC–MS analysis. An active compound solution was subjected to LC–ESI-QTOF-MS spectroscopy. The chemical compound was infused (without column) into a liquid chromatography (LC) coupled with an electrospray ionization quadrupole time-of-flight mass spectrometer (LC–ESI-QTOF-MS). The sample was placed in an Agilent 1260 Infinity Series HPLC System (Agilent Technologies, Waldbronn, Germany). The mobile phase was 0.1% (*v*/*v*) formic acid in water (A) and 0.1% (*v*/*v*) formic acid in acetonitrile (B) using an isocratic method at 50% (A). The flow rate was 0.5 mL/min, the injection volume was 1 μL, the column compartment was set at 35 °C, and the run time was 1 min. Mass detection was carried out with a 6540 Ultra High Definition Accurate Q-TOF-mass spectrometer (Agilent Technologies, Singapore). It was operated with electrospray ionization (ESI) in the positive ion mode in the *m*/*z* range of 100–1000 amu. The mass spectrometric conditions were set as follows: drying nitrogen gas (N_2_) flow rate of 10 L/min, gas temperature of 350 °C, nebulizer gas pressure of 30 psi, capillary voltage of 3.5 kV, fragment potentials of 100 V, skimmer of 65 V, Vcap of 3500 V, and Octopole RFP of 750 V. All mass acquisition and analysis were performed using Agilent MassHunter Data Acquisition Software version B.05.01 and Agilent MassHunter Qualitative Analysis Software B 06.0, respectively (Agilent Technologies, Santa Clara, CA, USA). Agilent calibration standard References A and B for MS were used to calibrate and tune the system before use. The peak was analyzed and identified by matching the mass spectra with the mass bank database.

### 4.9. Statistical Analysis

Statistical analyses were performed using GraphPad Prism 8 software. Data are presented as the mean ± SD of three replicates. All data were checked for normal distribution. The statistical difference in the mean zone of inhibition of PJ85 for individual test bacterium was carried out by using a one-way analysis of variance (ANOVA) followed by Fisher’s post hoc LSD test. Significant differences between incubation temperature and period were compared using a two-way ANOVA followed by Bonferroni correction. Significant differences in the maximum antibacterial activity between 30 °C and 37 °C were compared using Student’s *t*-test. A *p*-value < 0.05 denoted the presence of a statistically significant difference.

## 5. Conclusions

In the present study, *Streptomyces* sp. PJ85 was isolated from forest soil in Thailand and identified as *S. actinomycinicus*. The extracellular metabolites produced by *Streptomyces* sp. PJ85 exhibited a narrow spectrum of antibacterial activity against Gram-positive bacteria, including MRSA. Two bioactive components of PJ85 that were isolated and described based on LC–MS analysis were identified as actinomycin D, the primary compound, and DGLA, the minor compound. In this regard, it should be highlighted that this is the first report on the isolation and identification of bioactive substances from *S. actinomycinicus*. Evidence on the production of DGLA from *S. actinomycinicus* was also presented for the first time. However, future studies should include NMR characterization to validate the structure of PJ85_F39.

## Figures and Tables

**Figure 1 antibiotics-11-01797-f001:**
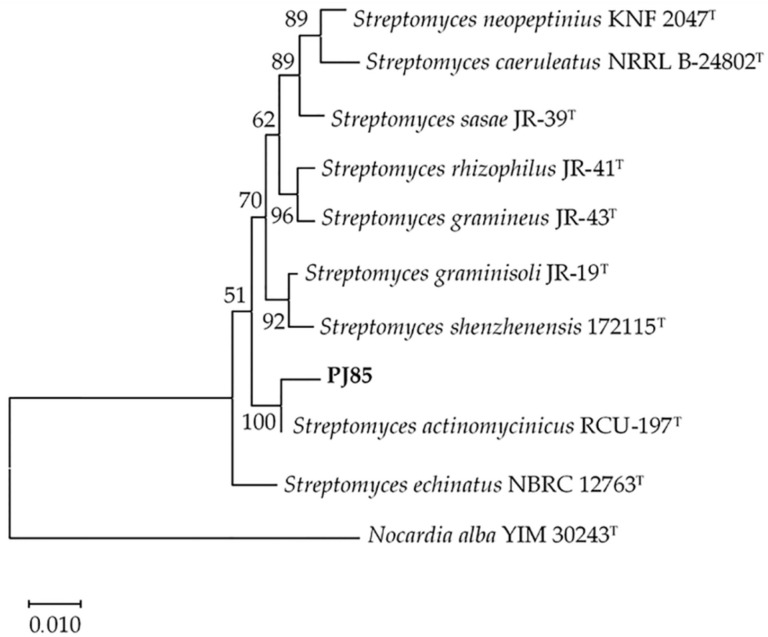
Neighbor-joining phylogenetic tree based on relative 16S rRNA gene sequences showing the phylogenetic relationships among *Streptomyces* sp. PJ85 and its closest strains. Numbers at nodes indicate bootstrap percentages (1000 replicates), and only values greater than 50% are shown at the nodes. The scale bar indicates 0.01 substitutions per nucleotide position.

**Figure 2 antibiotics-11-01797-f002:**
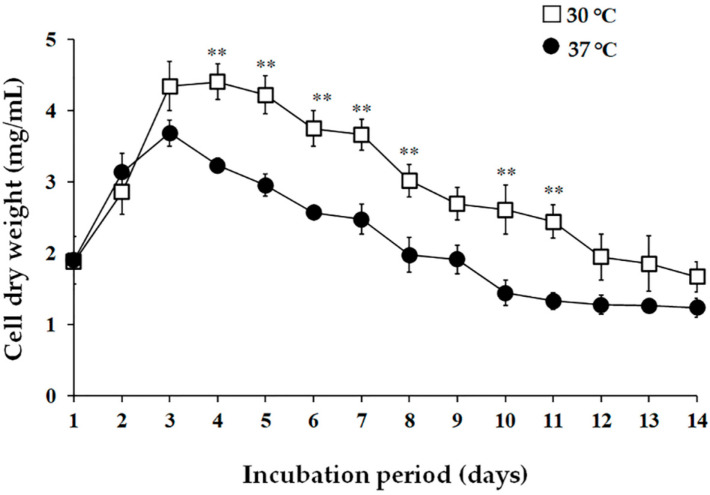
Effect of incubation temperature on the growth of *Streptomyces* sp. PJ85 cultured in ISP-2 medium. Incubation temperatures: (□) 30 °C and (●) 37 °C. Error bars represent the standard deviation of the mean (*n* = 3). The statistical significance of the differences between the cell biomass of PJ85 cultured at 30 °C and 37 °C was estimated using a two-way ANOVA with a Bonferroni multiple comparison test, ** *p* < 0.0001, compared on the same incubation day.

**Figure 3 antibiotics-11-01797-f003:**
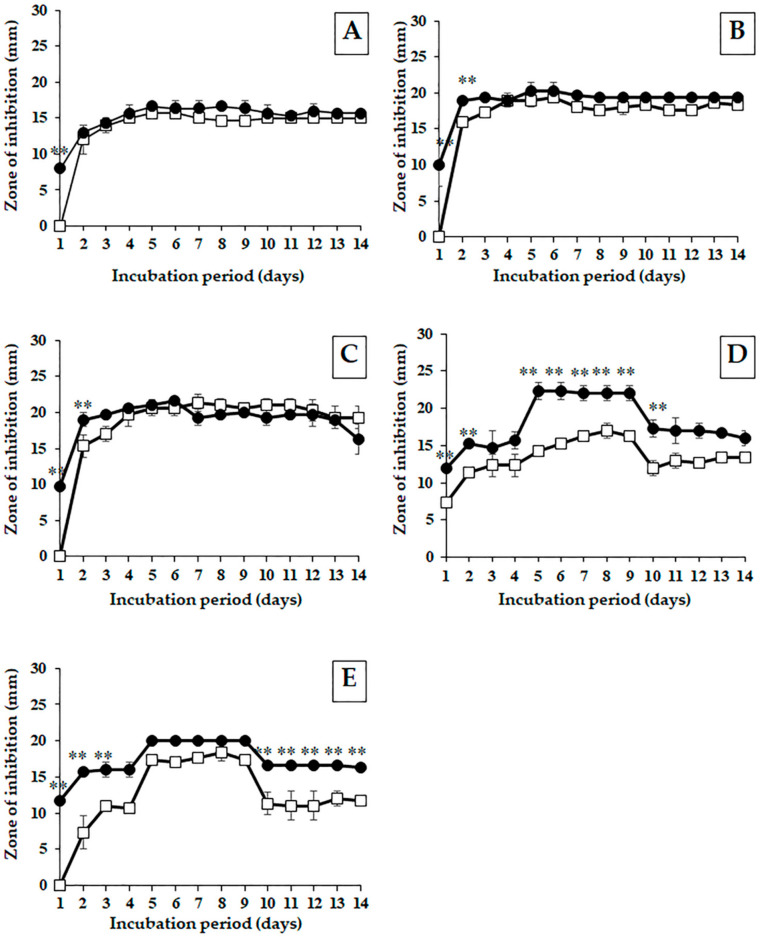
Effect of incubation temperature on the activity of antibacterial agents of *Streptomyces* sp. PJ85 cultured in ISP-2 medium against (**A**) MRSA DMST20651, (**B**) *S. aureus* ATCC29213, (**C**) *S. epidermidis* TISTR518, (**D**) *B. subtilis* TISTR008, and (**E**) *B. cereus* TISTR687. Incubation temperatures: (□) 30 °C and (●) 37 °C. Error bars represent the standard deviation of the mean (*n* = 3). The statistical significance of the differences between antibacterial activity of the crude extract of PJ85 cultured at 30 °C and 37 °C was determined using a two-way ANOVA with a Bonferroni multiple comparison test, ** *p* < 0.0001, compared on the same incubation day.

**Figure 4 antibiotics-11-01797-f004:**
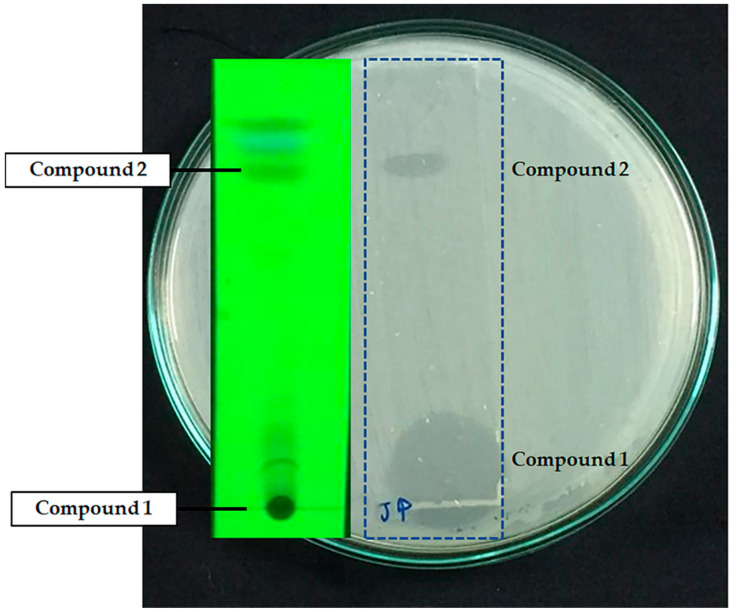
TLC-based bioautography of the ethyl acetate crude extract of PJ85 exhibiting antibacterial activity against MRSA DMST20651.

**Figure 5 antibiotics-11-01797-f005:**
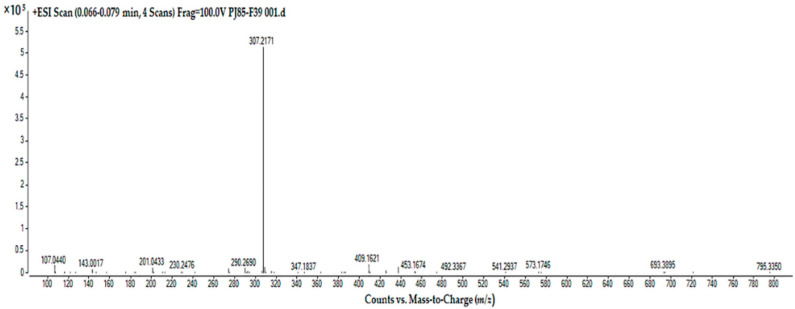
Liquid chromatography–mass spectrometry (LC–MS) analysis of compound PJ85_F39 in positive ion mode. MS spectrum showing the ion clusters for [M + H]^+^ at *m*/*z* 307.2171 correlated to dihomo-γ-linolenic acid (DGLA).

**Table 1 antibiotics-11-01797-t001:** Antibacterial activity of *Streptomyces* sp. PJ85 against test pathogens according to the perpendicular streak method.

Test Pathogens	Zone of Inhibition (mm)
MRSA DMST20651	50.00 ± 0.00 ^a^
*S. aureus* ATCC29213	46.67 ± 0.58 ^ab^
*S. epidermidis* TISTR518	48.33 ± 2.89 ^ab^
*B. subtilis* TISTR008	45.00 ± 3.00 ^b^
*B. cereus* TISTR687	38.33 ± 1.15 ^c^

Data are presented as mean ± standard deviation (*n* = 3); ^a, b^ and ^c^: different letters represent significance (LSD, *p*-value < 0.05).

**Table 2 antibiotics-11-01797-t002:** Inhibition zone diameter of crude ethyl acetate of *Streptomyces* sp. PJ85 collected at day 5 of cultivation.

Test Microorganisms	Zone of Inhibition (mm)		*p*-Value
	At 30 °C	At 37 °C	
MRSA DMST20651	15.67 ± 0.58	16.67 ± 0.58	0.259
*S*. *aureus* ATCC29213	19.33 ± 0.58	20.33 ± 1.15	0.188
*S*. *epidermidis* TISTR518	21.33 ± 1.15	21.67 ± 0.58	0.757
*B*. *subtilis* TISTR008	17.00 ± 1.00	22.33 ± 1.15 *	0.001
*B*. *cereus* TISTR687	18.33 ± 1.15	20.00 ± 0.00 *	0.003

Each value represents the mean ± SD of three independent experiments. *: at a confidence level of 95%, *p*-value < 0.05 is acceptable.

**Table 3 antibiotics-11-01797-t003:** The MIC values of *Streptomyces* sp. PJ85 crude ethyl acetate against test pathogens.

Test Pathogens	Minimum Inhibitory Concentration (MIC) (μg/mL)			
	Crude Compounds	Oxacillin	Vancomycin	Tetracycline
MRSA DMST20651	2	512	2	32
*S. aureus* ATCC29213	2	0.125	1	0.5
*S. epidermidis* TISTR518	16	0.125	2	0.0625
*B. subtilis* TISTR008	2	0.25	0.5	4
*B. cereus* TISTR687	1	64	2	0.0312

Values are the mean of *n* = 3 experiments.

**Table 4 antibiotics-11-01797-t004:** The nearest chemical compounds when matched to MassBank database detected in compound PJ85_F39 of *Streptomyces* sp. PJ85.

Name of the Compound	Chemical Formula	Molecular Weight (g/mol)	Sources	Properties	References
Dihomo-γ-linolenic acid (DGLA)	C_20_H_34_O_2_	306.50	*Mortierella* spp.	Antibacterial	[33]
Epigallocatechin	C_15_H_14_O_7_	306.27	*Camellia sinensis*	Antioxidant	[34]
2,3-*trans*-3,4-*cis*-leucocyanidin	C_15_H_14_O_7_	306.27	*Plantain banana*	Antiulcerogenic	[35]
Eremofortin A	C_17_H_22_O_5_	306.35	*Penicillium roqueforti*	Mycotoxin	[36]
Fenazaquin	C_20_H_22_N_20_	306.40	Synthetic chemical	Acaricide/insecticide	[37]
Feruloyl agmatine	C_15_H_22_N_4_0_3_	306.36	*Triticum aestivum* L. cv Chihokukomugi	Antifungal	[38]
Fluconazole	C_13_H_12_F_2_N_60_	306.10	Synthetic chemical	Antifungal	[39]
Koumine	C_20_H_22_N_20_	306.40	*Gelsemium*	Anti-inflammatory, analgesic and neurosteroid-modulating	[40]

## Data Availability

Not applicable.

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
