# Peer review of "Isolation and Identification of Bioactive Compounds from Streptomyces actinomycinicus PJ85 and Their In Vitro Antimicrobial Activities against Methicillin-Resistant Staphylococcus aureus"

_antibiotics, 2022, doi:10.3390/antibiotics11121797_

Round 1

Reviewer 1 Report

It is very well-marked that this study is acceptable with minor revision and useful for publish in this journal.

Author Response

" Please see the attachment "

Reviewer 2 Report

This paper would be certainly interesting if new compounds had been undoubtedly identified. However, al of the conclusions are not sufficiently supported by the results; the identification of the compounds by just an ion’s m/z is not enough. While I understand that the laboratories involved in this research could have limitations of funding and resources, I am afraid I do not think this manuscript is fit for publication.

Main concerns:

Compounds have not been identified, they are merely possibilities based uniquely on an MS m/z value. It has also not been demonstrated that the ions chosen are the actual molecules responsible for the antibiotic activity. Having the facility of a high resolution LC-MS I would expect at the very least to run a proper HPLC chromatography on extracts or active fractions together with standards of the proposed compounds; comparison of retention times, MS and fragmentation patterns would strongly support the proposed identification. Even the same TLC and bioassay could be performed with standards. Or samples from antibiotic-producing conditions could be compared with samples from non-producing conditions. Of course, full purification and NMR structural elucidation would be the desirable objective.

The strain is assigned to a species by only 16 rDNA partial sequence.

The MS databased used is not very comprehensive, definitely not for natural products. In fact, I cannot find actinomycin in any way I search for it, by name, formula or mass.

Other points:

Figure 4: what is marked as compound 1 seems to me more like the origin where sample was spotted. Also, was the plate not stained at all, just UV exposure as stated in methods?

It is stated that “3 mg of compound 2” was obtained from the TLC (line 222) but that seems to me as a really large amount.

I find the need to be more precise with the terminology when referring to MS data, in particular the use of “accurate mass”, “monoisotopic mass”, ”molecular weight”. In fact, the molecular weight given in line 236 looks more like the monoisotopic mass, i.e. that of the molecule containing only the most common isotope of each element, and it should be properly calculated and given in Da. The standard procedure then is to use the monoisotopic mass to calculate likely molecular formulae and look for known compounds that could match. On this point, the manuscript never states any molecular formula calculated out of the MS data.

On lines 216-217 and 308-309 it is stated that S.actinomycinicus is known to produce actinomycin D but on lines 272-273 it is stated the opposite.

Lines 278-281: the strain can be useful for industrial fermentation only if it produces interesting compounds, which does not seem to be the case. As heterologous host capable of growing and producing at higher temperature it would have a very long way to go. Besides, the problem could be the opposite, having to heat up several thousand litres of medium is also not cheap.

Lines 283-285, these are results, not discussion, and the speculation about polarity is unnecessary,

Penicillin was not the first antibiotic being discovered. e.g. the synthetic chemical salvarsan (https://en.wikipedia.org/wiki/Arsphenamine)

Methods are very poor in many important aspects: description of the LC-MS methodology, column, gradient, MS set up, calibration, etc. Culture conditions, shaking, flasks with or without baffles (and in line 190 I find it difficult to believe a 1l flask can be filled with nearly half the volume of medium)

English language is good and the manuscript fully understandable, but there are many points in which an editorial revision is needed.

Author Response

" Please see the attachment "

Reviewer 3 Report

The manuscript entitled “Identification and antibacterial activity of new compound of Streptomyces actinomycinicus PJ85 isolated from dry dipterocarp forest soil in Northeast Thailand” by Panjamaphon Chanthasena et al., describes the isolation of Streptomyces PJ85 from forest soil in Nakhon Ratchasima, Thailand. The strain was identified by molecular techniques as S. actinomycinicus, which was evaluated for its antibacterial activity against Gram-positive and Gram-negative bacteria. The authors attributed the showed activity to two compounds identified by CG-EM.

Discussion on the isolation and identifying of the strain and the antibacterial activity is understandable, and the paper in general is reasonably written. Nevertheless, I would recommend being revised, and improved by an English speaker.

The results are interesting, however, in my opinion the active compounds are not properly identifying. Their structures should be established by spectroscopic means; therefore, I suggest isolating and identify the two compounds by spectroscopic means as Nuclear Magnetic Resonance.

I have some other comments that I would like the authors to address:

1.- Line 31, mass spectroscopy; must be mass spectrometry.

2.- Line 222, about 3.0 mg of compound 2; must be the weigh of the chromatographed extract or fraction.

3.- Line 230, crude ethyl acetate; must be ethyl acetate crude extract of...

4.- Lines 338-340. The numbers in molecular formulas must be in subscript.

Author Response

" Please see the attachment "

Reviewer 4 Report

The manuscript by Panjamaphon Chanthasena et al describes the isolation of single streptomycete strain and characterization of antibacterial compounds produced by this strain. While there is term "new compound" in the title of manuscript, in fact no new compound was detected. The manuscript is written in poor English with extensive use of laboratory jargon, missing words, many small errors, etc.

16S rRNA analysis is not sufficient for phylogenetic placement of Streptomyces spp. please use another marker(s) - e.g. gyrB, rpoA. Moreover, observed level of similarity ( 98.90%) is low an studied strains is clearly not a Streptomyces actinomycinicus strain (see length of branches in your tree).

Table 2. Inhibition zone diameter of crude ethyl acetate of Streptomyces???

Table 3. dtto

Possibly, the paper https://pubmed.ncbi.nlm.nih.gov/30217010/ should be discussed and cited

Author Response

" Please see the attachment "

Round 2

Reviewer 2 Report

I thank the authors for having taking the time for a detailed response, but I am afraid they have not resolved any of the main issues I raised and the manuscript has not changed in any substantial form. Furthermore, the strain and its antimicrobial activity had already been described in a previous paper cited as ref 26, and actinomycin D had already been shown to be produced by the similar strain S. actinomycinicus as the authors state in line 227 and ref 30, so the lines 504-506 are not really correct, even taking into consideration the Response 8. Summing up, if the strain is the same species and makes the same main well known active compound, the novelty of the work is even more diminished. I truly doubt about the identification of compound 2. I am sorry but I still consider this manuscript not suitable for publication, I still consider the data insufficient to support the conclusions, and overall a very insubstantial manuscript for publication.  I’m sorry, I provide below some suggestions for improving.

Additional comments:

- The fact that there might be other similar papers published in this or other journal (and I have not read them, but I doubt they have as little data as this manuscript) does not imply I agree with that or I should accept that this one is fit for publication, this is a decision for the editor to make. If you have more data or are close to obtain structural elucidation of compound 2 I would suggest to prepare a new manuscript with it, maybe together with a proper species description.

-Full genome sequencing is extremely affordable these days; I have used in the past the services of https://microbesng.com/ which, despite not providing a high quality assembly, the service is very cheap and provide sufficient data. With genome information available you can then address the species description based on services like https://tygs.dsmz.de/

-Run an actinomycin D standard under the same conditions to compare for retention time and observed spectrum; spike the sample with standard. This should provide a more substantial proof for your conclusions.

-The addition of house-keeping gene sequence results to the discussion (lines 281-290) without even mentioning it in Results and Methods, and submitting the sequences to public repositories or adding them in SI, is an unacceptable practice.

-A more detailed LC-MS result can easily be provided, with spectra zoomed around the actual monoisotopic ion to see the full isotopic pattern. This is a critical information for compound identification based on LC-MS

Reviewer 4 Report

Revised version of manuscript is clearly improved, however, I still see some discrepancies and small errors

Lane 83 - according GenBank data the sequence length is 1473 bp - not 1523 bp as shown in the manuscript

Lane 231 - scrapped

Lane 287 - (gyrB; 98.99%, rpoA; 99.51%, atpD; 99.58%, and rpoB; 99.08%) - you are discussing data not shown in the manuscript and unavailable in GenBank database

Lane 376 - sanger - Sanger (dideoxy) 
